# Tuning Bandgaps of Mixed Halide and Oxide Perovskites CsSnX$_3$ (X=Cl, I), and SrBO$_3$ (B=Rh, Ti)

Hongzhe Wen and Xuan Luo *

National Graphene Research and Development Center, Springfield, VA 22151, USA; hongzhewen01@gmail.com
* Correspondence: xluo@ngrd.org

**Abstract:** Perovskites have recently attracted interest in the field of solar energy due to their excellent photovoltaic properties. We herein present a new approach to the composition of lead free perovskites via mixing of halide and oxide perovskites that share the cubic ABX$_3$ structure. Using first-principles calculations through Density Functional Theory, we systematically investigated the atomic and electronic structures of mixed perovskite compounds composed of four cubic ABX$_3$ perovskites. Our result shows that the B and X atoms play important roles in their band structure. On the other hand, their valence bands contributed by O-2p, Rh-4p, and Ti-3p orbitals, and their electronic properties were determined by Rh-O and Ti-O bonds. With new understandings of the electronic properties of cubic halide or oxide perovskites, we lastly combined the cubic perovskites in various configurations to improve stability and tune the bandgap to values desirable for photovoltaic cell applications. Our investigations suggest that the mixed perovskite compound Cs$_2$Sn$_2$Cl$_3$I$_3$Sr$_2$TiRhO$_6$ produced a bandgap of 1.2 eV, which falls into the ideal range of 1.0 to 1.7 eV, indicating high photo-conversion efficiency and showing promise towards solar energy applications.

**Keywords:** perovskites; solar energy; electronic structure





## 1. Introduction

In response to the increasing demand for solar cells as the source of renewable energy, photovoltaic solar panels based on perovskite semiconductors have been gaining wide attention [1–5]. Perovskite is a calcium titanium oxide mineral, discovered in the Ural Mountains of Russia, that exhibited superior transmission of solar energy compared to silicon or selenium, as high as 27% [6–8]. However, the instability of lead within perovskite minerals has introduced new environmental concerns, as temperature or humidity fluctuations can cause toxic lead-based substances to leak into the environment [9–13].

For applications in semiconductors for solar cells, perovskites require strong electronic properties which can be understood from their band structures and bandgap values. Although cubic perovskites possess bandgaps that are suitable for many applications [14,15], the bandgap values are often inadequate for solar cells, and may poorly bond with other components during use. To resolve issues in stability and transmission efficiency, researchers have looked towards halide perovskites, hybrid perovskites, and double perovskites, yet the presence of lead or other toxic components generates environmental concerns [16–22]. Thus, the search for lead free, nontoxic, and inorganic perovskite materials is an urgent and ongoing research field [23,24].

We herein present a new approach to the composition of lead-free perovskites via mixing of halide and oxide perovskites, enabling tuning of structural stability and bandgap values to design effective photovoltaic materials [25]. Specifically, we mixed, combined, stretched, and compressed various cubic perovskites symmetrically to modify the structural and electronic properties, furthering our understanding of mixed perovskite compounds and improving transmission efficiency [26,27]. To form the mixed perovskite compound, we mixed halide or oxide perovskites that share the same ABX3 structure, where A is a cation with different groups of valence electron configurations, B is a cation chosen from transition

metals, and X is either a halide or an oxide [28]. In particular, we selected $CsSnCl_3$, $CsSnI_3$, $SrRhO_3$, and $SrTiO_3$ as exceptional halide or oxide perovskites as they exhibit similar electronic properties. Moreover, halide perovskites can bond strongly to other perovskites within thin films [29]. However, this combination approach provided challenges as perfect cubic symmetry is rather uncommon [30–33], while the actual bandgap produced by a single cubic perovskite is not theoretically applicable. Lastly, we constructed the mixed perovskite compounds in diverse arrangements and orders to tune the computational bandgap to values ideal for solar cell applications.

## 2. Calculation Methods

### 2.1. Computational Details

All first-principle calculations performed in this research were completed within the framework of the Density Functional Theory (DFT) as implemented in the ABINIT [34,35] code. The Generalized Gradient Approximation (GGA) [36] in the Perdew-Burke-Ernzerhof (PBE) [36] form was selected for the Exchange Correlation (XC) functional. For all elements in our calculations, we used the Projected Augmented Wave (PAW) [37] pseudopotentials with projectors generated with the AtomPAW [38–40] code. GGA-PBE DFT calculations tend to contain the issue of underestimated bandgaps [41]. As a result, the underestimation may appear in the final result of our calculation, and may therefore suggest higher efficiency than the bandgaps had when tuning the bandgap of mixed perovskites. In light of this and in order to render accurate data, we use Equation (1) to adjust the final value.

$$E_{gap}(corrected) = (E_{gap}(GGA - PBE) \times 1.358 + 0.125) \times 0.998 + 0.014 \quad (1)$$

This equation is necessary in order to make comparison between actual material and theoretical calculation [42]. The displayed equation is adjusted to fit the difference between different types of perovskites. The corrected value gives similar result as the experimental values mentioned in Table 1.

**Table 1.** Calculated lattice parameter (a) and previous calculations as well as experimental data.

| Compounds | a (Bohr) | Experimental Data (Bohr) | Error |
|-----------|----------|--------------------------|-------|
| $CsSnCl_3$ | 10.639 | 10.609 [42] | 0.282% |
| $CsSnI_3$ | 11.856 | 11.826 [42] | 0.253% |
| $SrRhO_3$ | 7.522 | 7.427 [43] | 1.263% |
| $SrTiO_3$ | 7.441 | 7.446 [44] | 0.067% |

### 2.2. Materials

As the basic unit cells, we selected from cubic perovskites including $CsSnCl_3$, $CsSnI_3$, $SrTiO_3$, and $SrRhO_3$. To perform computational studies, we first optimized each of the halide or oxide perovskites individually. To investigate the structural and electronic properties of mixed perovskite compounds, we then constructed $2 \times 2 \times 2$ supercells from various configurations of the four cubic perovskites. By studying the effects of various perovskite mixtures on the overall stability and bandgap, we sought to achieve values that were optimal for photovoltaic cell applications.

We performed computations based on PAW pseudopotentials based on the electronic configurations of the elements as displayed in Table 2.

**Table 2.** Electron configurations and radius cutoff for generate PAW pseudopotentials atom.

| Atom | Electron Configurations | Radius Cutoff (Bohr) |
|------|------------------------|---------------------|
| O | [He] $2s^2 2p^4$ | 1.4147 |
| Cl | [Ne] $3s^2 3p^5$ | 1.8032 |
| Ca | [Ar] $4s^2$ | 1.9142 |
| Ti | [Ar] $3d^2 4s^2$ | 2.3000 |
| Sr | [Kr] $5s^2$ | 2.2067 |
| Rh | [Kr] $4d^8 5s^1$ | 2.4031 |
| Sn | [Kr] $4d^{10} 5s^2 5p^2$ | 2.5109 |
| I | [Kr] $4d^{10} 5s^2 5p^5$ | 2.3002 |
| Cs | [Xe] $6s^1$ | 2.2066 |

## 2.3. Convergence Calculations

To perform convergence calculations for the cubic unit cells, we conducted kinetic energy cut off and k-point mesh converged for the $abx_3$ perovskite materials. We performed self-consistent field (scf) cycles until the total energy reached a difference of $1 \times 10^{-10}$ hartree. We performed these convergence calculations using experimental lattice constants. All variables were considered to be in convergence when the energy difference reached below 0.0001 hartree twice. Then, we determined the fully relaxed parameters and interval coordinates when the maximum dilation was 1.05. These values will be considered in later calculations. Kinetic energy cutoff and k-point mesh convergence of four compounds are displayed in Table 3.

**Table 3.** Kinetic energy cut off (ecut), k-point mesh convergence for cubic perovskites.

| Compounds | Ecut (ha) | K Point Mesh |
|-----------|-----------|--------------|
| $Cssncl_3$ | 32.0 | 6 6 6 |
| $Scsni_3$ | 20.0 | 6 6 6 |
| $Srrho_3$ | 20.0 | 6 6 6 |
| $Srtio_3$ | 20.0 | 4 4 4 |

We then constructed an inorganic cubic $2 \times 2 \times 1$ perovskites supercell ($abx_3$) by combining two halide and two oxide perovskites. We sought to design theoretical mixed perovskite compounds that showed advantageous properties as solar cell materials, including cost-efficiency, structural stability, and ideal bandgap values, as well as to better understand the effects of mixed halide and oxide perovskites on the electronic band structure and bandgap towards photovoltaic applications.

## 2.4. Band Structures Calculation

We used the relaxed atomic structures shown above to calculate each perovskites band structure. We calculated the band structures along four different high symmetry k-points in the Brillouin Zone, Γ (0,0,0), R (0.5, 0.5, 0.5), X (0, 0.5, 0), and M (0.5, 0.5, 0). Each valence band represents one electron, and we added a total of four conduction bands. The electronic band structure is used to determine the electronic properties of perovskites. We transferred the lattice vectors to reciprocal spaces by using the formula below:

$$\vec{b_1} = 2\pi \frac{a_2 \times a_3}{a_1 \cdot (a_2 \times a_3)} \tag{2}$$

$$\vec{b_2} = 2\pi \frac{a_3 \times a_1}{a_2 \cdot (a_3 \times a_1)} \tag{3}$$

$$\vec{b_3} = 2\pi \frac{a_1 \times a_2}{a_3 \cdot (a_1 \times a_2)} \tag{4}$$

### 2.5. Formation Energy

We calculated formation energies of our mixed perovskites to investigate the stability of our compounds. We computed the total energies of the single cubic perovskites as well as the $2 \times 2 \times 1$ mixed compounds. The formation energy was calculated using the formula below:

$$E_f = E_{mixed} - E_1 - E_1 - E_3 - E_4 \tag{5}$$

where $E_{mixed}$ represents the total energy of $2 \times 2 \times 1$ mixed perovskites, $E_1$, $E_2$, $E_3$ and $E_4$ represent the energies of the individual cubic perovskites that compose the mixed perovskite compound.

## 3. General Results and Discussion

### 3.1. Structural Properties

We chose all four cubic perovskites to possess regular cubic symmetry. The atoms of primary unit cells were positioned at A (0, 0, 0), B (0.5, 0.5, 0.5), $X_1$ (0.5, 0.5, 0), $X_2$ (0.5, 0, 0.5) and $X_3$ (0.5, 0, 0.5). We relaxed lattice parameters including four cubic perovskites and several $2 \times 2 \times 1$ mixed perovskite combinations. The lattice parameters of cubic perovskites were in agreement with previous literature [42–44]. The equilibrium lattice parameters (a) are displayed in Table 1. The convergence calculations and calculated lattice parameters revealed that for halide perovskites, smaller halogens resulted in much shorter and stronger bonds compared to larger halogens (Tables 1 and 2). For example, since Cl has an atomic radius of 1.8032 Bohr and I has an atomic radius of 2.3002 Bohr (Table 2), the Sn-Cl bond is stronger than Sn-I bonds, improving the efficiency of the semiconductor while utilizing cost-efficient materials.

### 3.2. Pure Materials

To study the electronic properties of the mixed perovskite compounds, we performed band structure studies, total density of states (TDOS) plots, projected density of states (PDOS) plots, as well as fat band analysis at zero pressure. We first individually investigated the four basic compounds that would form mixed $2 \times 2 \times 1$ cubic perovskites, namely $CsSnCl_3$, $CsSnI_3$, $SrRhO_3$, and $SrTiO_3$. These valuable research materials are cost-efficient and effective in both financial and environmental terms. Thus, the resulting mixed compound would be composed of various configurations of halide and oxide cubic perovskites, enabling the bandgap to be tuned to desirable values. We calculated kinetic energy cutoff, k-points mesh and performed band structures, total density of states (TDOS) and projective density of states (PDOS) using GGA-PBE DFT calculations along high symmetry directions in the First Brillouin zone with relaxed but symmetrical structures.

The band structure analysis provided insight into the electronic properties and electronic bands of the cubic perovskites as promising semiconductors. Based on the relaxed parameters and interval coordinates, we respectively calculated band structures and performed fat band calculations along the high symmetry points located at Γ (0, 0, 0), R (0.5, 0.5, 0.5), X (0, 0.5, 0), and M (0.5, 0.5, 0). Figure 1 exhibits the basic structure of a perovskite unit cell and the Brillouin Zone. The Fermi level for all four perovskites were set to zero. Our bandgaps were close to previously reported experimental bandgaps. For the band structures figures, all the peaks are located on the R point (see Figure 2). All cubic perovskites, except for $SrRhO_3$, exhibited bandgaps on the DOS plots. DOS and PDOS plots are all presented in Figure 3.

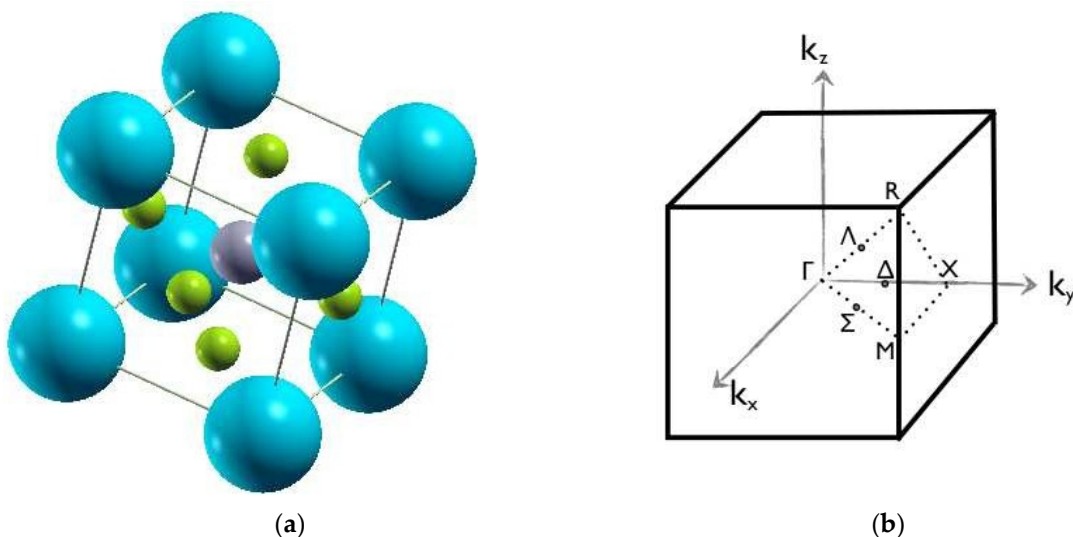

(a) (b)

**Figure 1.** (**a**) Cubic structure of formal perovskites. The spheres in the figure are the atoms. The gray atom in the center represents A atoms, the green atoms represent the B atoms, and the blue atoms are the X atoms. (**b**) First Brillouin zone of cubic crystal structure [42]. High symmetry k-points are Γ (0,0,0), R (0.5, 0.5, 0.5), X (0, 0.5,.0), and M (0.5, 0.5, 0).

### 3.2.1. CsSnX$_3$ (X=I, Cl)

By observing the band structures of CsSnX$_3$ groups and comparing the two halide perovskites, we determined that CsSnI$_3$ obtained a narrower bandgap than CsSnCl$_3$. The distance between conduction band minimum (CBM) and valence band maximum (VBM) was 1.1 eV to 0.4 eV. The bandgap, electronic properties and bonding intensity can all be tuned using various halogens. For example, the orbitals around CsSnCl$_3$'s Fermi level namely, the Sn-p and Cl-p orbitals which occupy the VBM and CBM respectively are distinctly positioned in the respective ranges of 0 eV to 7.0 eV and −2.0 eV to −7.0 eV. The PDOS plots of CsSnCl$_3$ revealed a Cl-p peak on −4.0 eV and an Sn-p peak on 3.5 eV. Similar to the orbitals of CsSnCl$_3$ above, the Sn-p and I-p orbitals of CsSnI$_3$ were distinctly occupied on VBM and CBM respectively: the former was positioned between 0.0 eV to 4.0 eV, while the latter was located between −1.0 eV to −4.0 eV.

As a comparison with CsSnCl$_3$, the peak appeared to be similar to the approach heights. These results show that the two halide perovskites, CsSnCl$_3$ and CsSnI$_3$, likely undergo hybridization due to Sn-halide bonding, although CsSnI$_3$ appears to have a stronger hybridization effect.

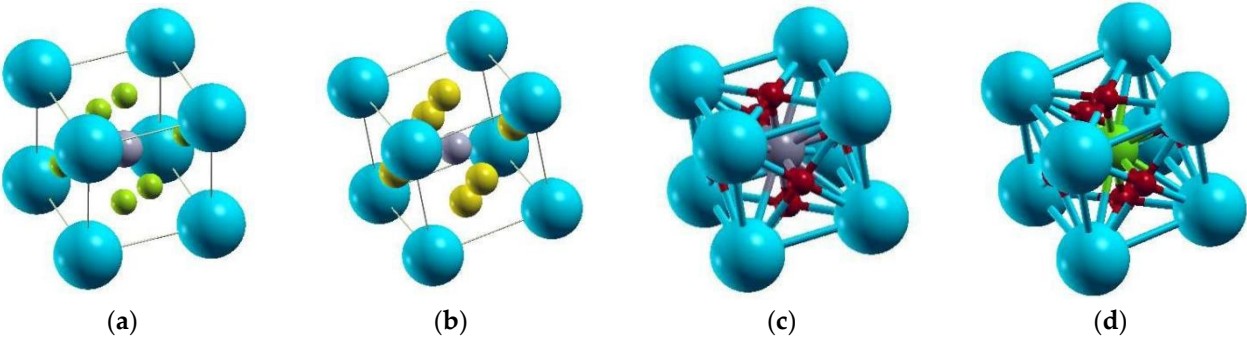

(a) (b) (c) (d)

**Figure 2.** *Cont.*

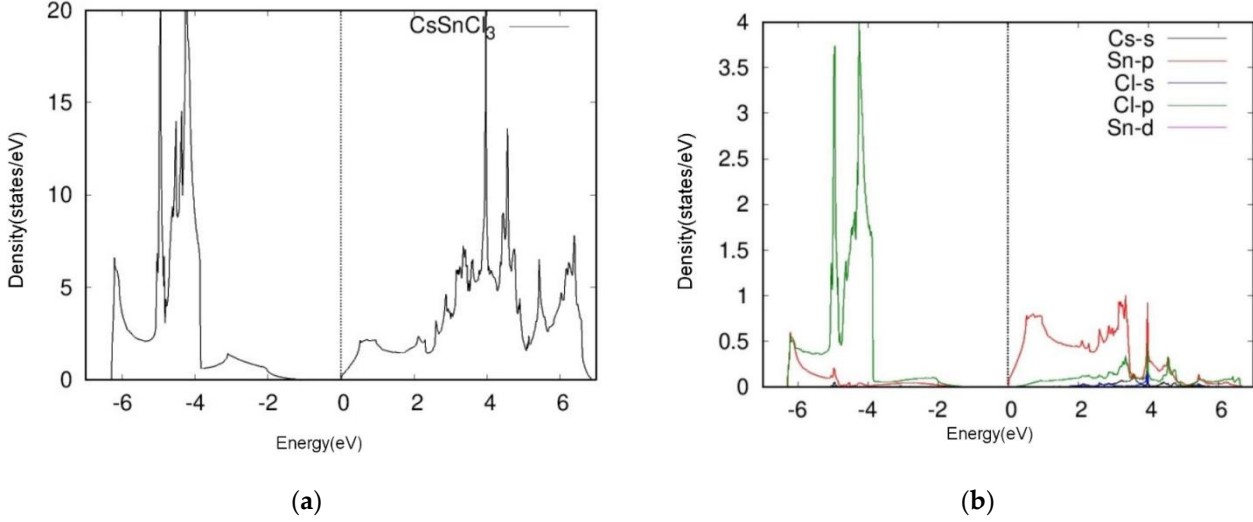

**Figure 2.** Electronic band structures of four cubic perovskites (**a**) CsSnCl₃, (**b**) CsSnI₃, (**c**) SrTiO₃, (**d**) SrRhO₃, (**e**) band structure of CsSnCl₃, (**f**) band structure of CsSnI₃. (**g**) band structure of SrTiO₃, (**h**) band structure of SrRhO₃. The Fermi level is set to 0 for all band structures.

**Figure 3.** *Cont.*

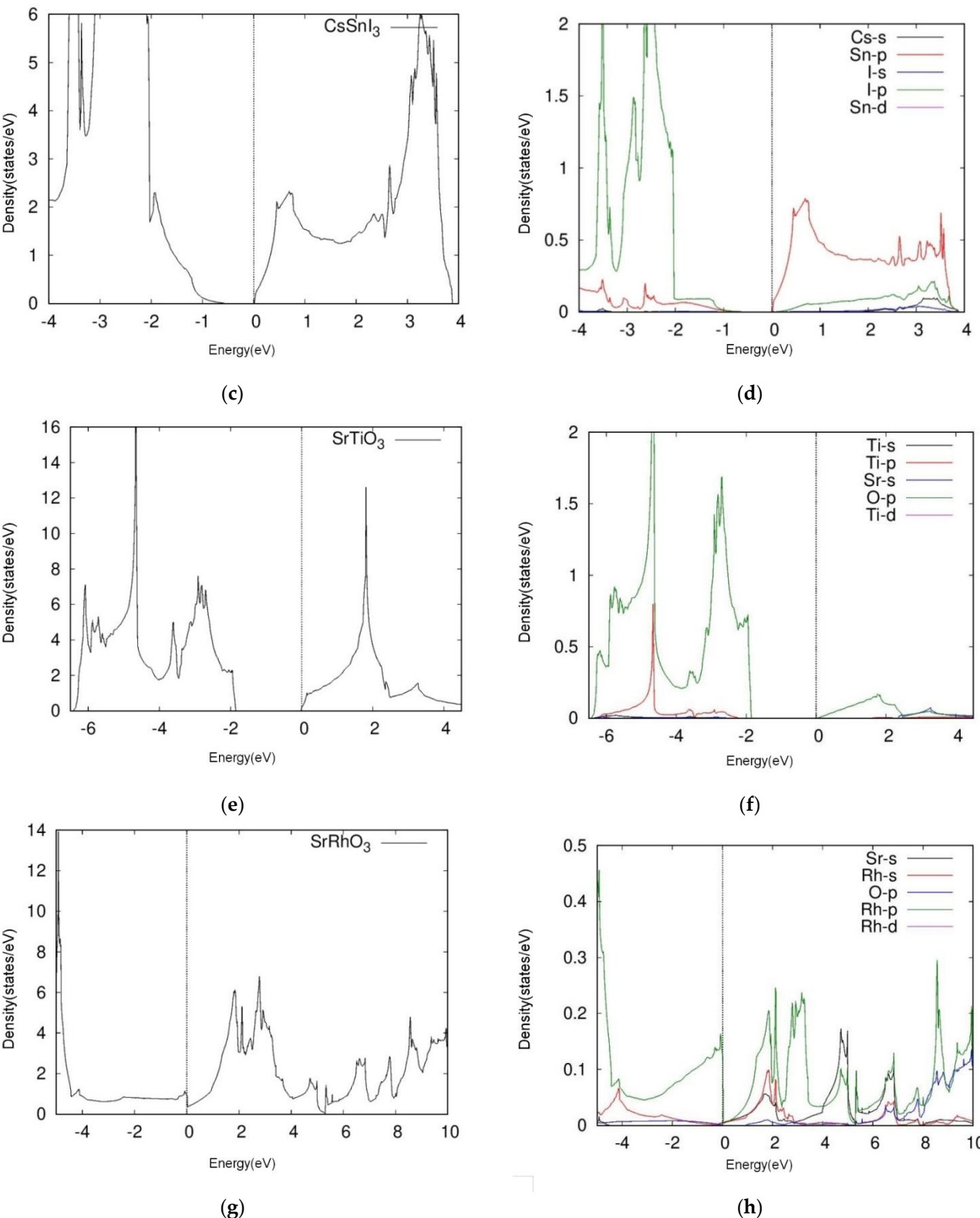

**Figure 3.** Total DOS (left) and projective (right) DOS of four kinds of cubic perovskites (**a,b**) and CsSnCl$_3$, (**c,d**) CsSnI$_3$, (**e,f**) SrTiO$_3$, (**g,h**) SrRhO$_3$. We calculated the A-s, B-p, B-d, X-s, and X-p orbitals of ABX$_3$ cubic perovskites.

### 3.2.2. SrBO$_3$ (B=Rh, Ti)

We next examined the two oxide perovskites, SrRhO$_3$ and SrTiO$_3$, which revealed completely different electronic properties and thus different band structures compared to those of the halide perovskites. First, oxide perovskites revealed indirect bandgaps through R-Γ, whereas the halide perovskites presented direct bandgaps. SrRhO$_3$ showed no bandgap according to its energy band figure, while SrTiO$_3$ had a bandgap of 1.8 eV. By analyzing TDOS and PDOS plots, we next determined that SrRhO$_3$ has a valence band that originates from O-p and Rh-d states. In fact, the Rh-d band, which is near the Fermi level, presented the greatest contribution to the VBM of SrRhO$_3$ (Figure 3h). Our computational results indicated that SrRhO$_3$ perovskites undergo p-p bonds, as the Rh-p orbitals predominated the electronic states (Figure 3h).

On the other hand, the VBM and the CBM of SrTiO$_3$ were both populated by O-2p orbitals. Moreover, the figure suggests hybridization between the two states. This phenomenon reveals unique characteristics of SrTiO$_3$ perovskites, particularly regarding Ti-O bonding, as the O-p orbital appears to share a peak with the Ti-p orbital near −4.9 eV (Figure 3f).

Following, we performed Fat band analysis to compare to our Density of States calculations, revealing that A cations likely reside on the top of the conduction bands, B cations likely stay within the VBM, while the three X cations tend to populate the vacancies between A and B cations. Therefore, the selection of halide and oxide perovskites appears to direct the order of the orbitals and placement within band structures. Figure 4 shows the fat band of halide and oxide perovskites.

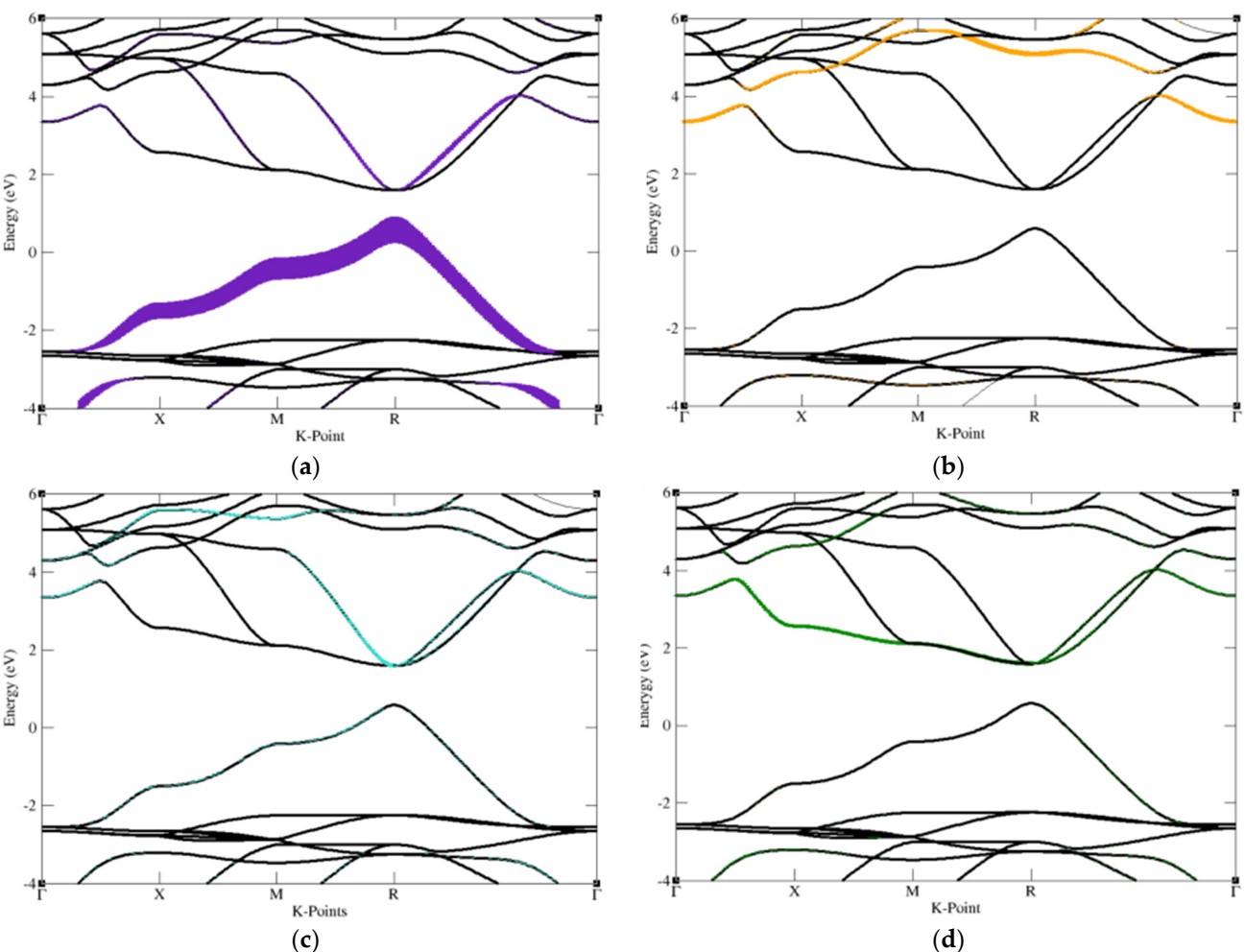

**Figure 4.** *Cont.*

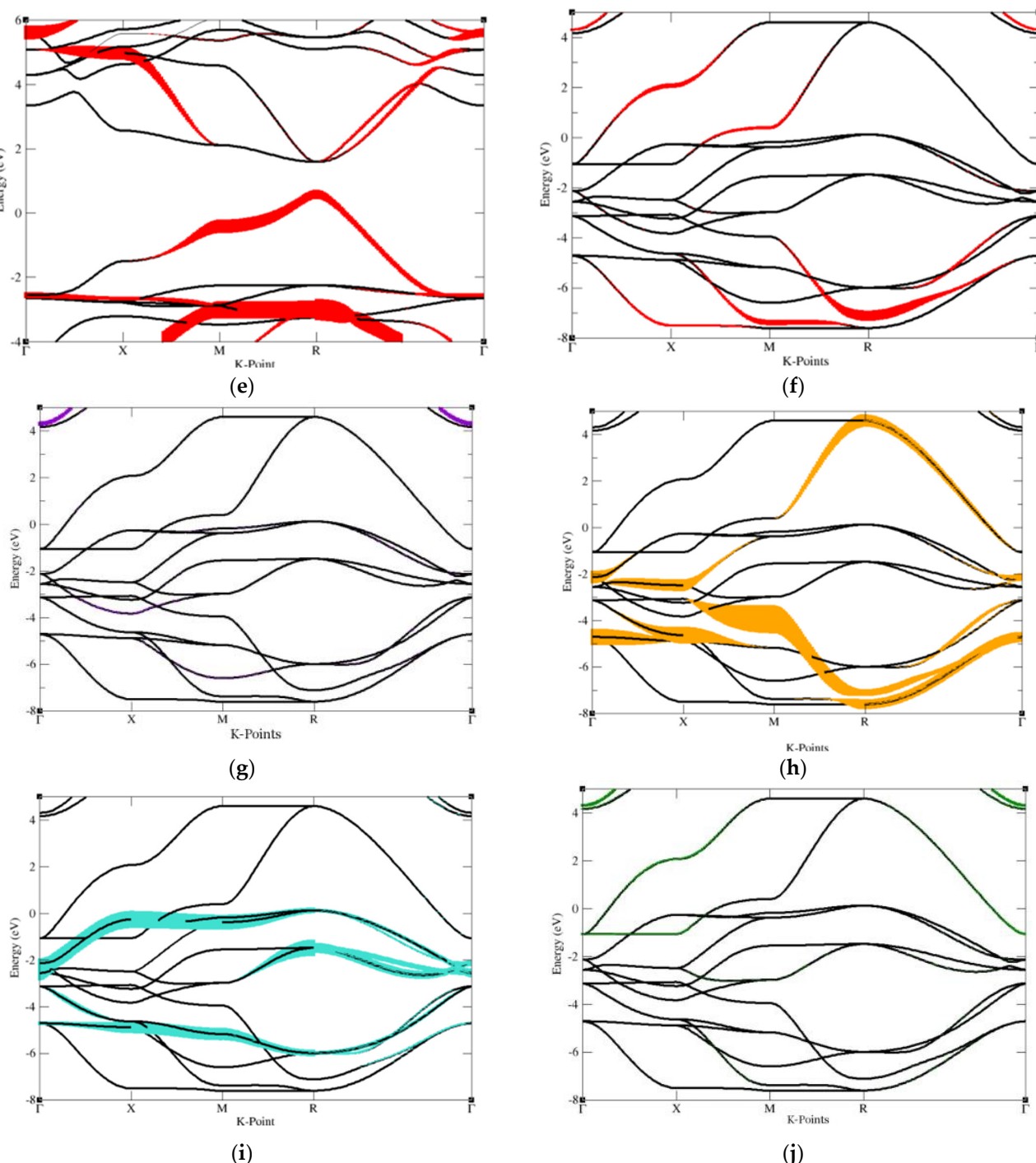

**Figure 4.** Fat band of halide and oxide perovskite, the main contribution of each atom contained in cubic perovskite structure graphed out separately. These graphs are drawn for the purpose of sufficiently observing the main contribution of the atoms contained in the structure in order to adjust the energy bandgap. (**a**) Cs atom contained in $CsSnCl_3$ (**b**) Sn atom contained in $CsSnCl_3$, (**c–e**) Cl atom contained in $CsSnCl_3$. (**f**) Sr atom of $SrRhO_3$, (**g**) Rh atom of $SrRhO_3$, (**h–j**) O atom of $SrRhO_3$.

### 3.3. Mixed Perovskites

With a better understanding of the structural and electronic properties of halide and oxide perovskites, we next sought to design mixed perovskite compounds to achieve the optimal bandgap ranges of 1.0 to 1.7 eV [44] for photovoltaic cell applications. We studied the effects of combining two oxide and two halide perovskites in various configurations,

while fixing the total number of atoms to 20 per supercell. We also computed the formation energies to obtain stability information on the mixed perovskite compounds. Table 4 displays the bandgap, corrected bandgap and formation energy of mixed perovskites.

**Table 4.** Electronic bandgap for $2 \times 2 \times 1$ mixed perovskites as well as formation energy.

| Compounds | Bandgaps (eV) | Corrected (eV) | Formation Energy (eV/atom) |
|---|---|---|---|
| $Cs_2Sn_2Cl_3I_3Sr_2TiRhO_6$ | 1.2 | 1.68 | 68.304 |
| $Cs_2Sn_2Cl_6Sr_2Ti_2O_6$ | 1.1 | 1.54 | 90.223 |
| $Cs_4Sn_4Cl_6I_6$ | 1.9 | 2.66 | 0.023 |
| $CsSnCl_3Sr_3Ti_2RhO_9$ | 1.05 | 1.47 | 34.4 |

### 3.3.1. $Cs_4Sn_4Cl_6I_6$

We hypothesized that mixing half of each of the halide perovskites would be the most efficient method of adjusting the bandgap to a desirable value, since both perovskites exhibit direct bandgaps and already possess bandgaps close to the optimal values. We tested mixing 50% of both halide perovskites, $CsSnCl_3$ and $CsSnI_3$, to form $Cs_4Sn_4Cl_6I_6$. The computational results showed an indirect bandgap on the $\Gamma$ point at approximately 1.9 eV, which was not in the range of the optimal values. Still, the formation energy was about 0.023 eV, indicating that the mixed halide perovskite formed a relatively stable structure, and that fewer defects may form under different concentrations or temperatures.

### 3.3.2. $Cs_2Sn_2Cl_3I_3Sr_2TiRhO_6$

We next combined two halide ($CsSnCl_3$, $CsSnI_3$) and oxide ($SrRhO_3$, $SrTiO_3$) perovskites to form the mixed compound, $Cs_2Sn_2Cl_3I_3Sr_2TiRhO_6$. The bandgap was located right on the $\Gamma$ point, with a value of approximately 1.2 eV. Since we used the GGA approximation, the experimental bandgap is likely to fall into the range of 1.0–1.7 eV. However, the formation energy was found to be 68.304 eV, indicating that under low concentrations or low temperature, defects may occur due to the structural instability of the compound. Such defects may also decrease the lifetime and efficiency of the semiconductor

### 3.3.3. $Cs_2Sn_2Cl_6Sr_2Ti_2O_6$

Furthermore, we combined two $CsSnCl_3$ and two $SrTiO_3$ perovskites to form $Cs_2Sn_2Cl_6Sr_2Ti_2O_6$, resulting in an indirect bandgap of 1.1 eV. We found that the VBM was located on the R point while the CBM was on the $\Gamma$ point. Due to the GGA approximation, the bandgap is likely to be in the optimal ranges for photovoltaic applications. However, the formation energy was very high, similar to the mixed compound $Cs_2Sn_2Cl_3I_3Sr_2TiRhO_6$ above, indicating the likelihood of defects at different concentrations or temperatures. We also found less electrons occupying under the Fermi level, and more gathered around the conduction bands.

### 3.3.4. $CsSnCl_3Sr_3Ti_2RhO_{12}$

Lastly, $CsSnCl_3Sr_3Ti_2RhO_{12,}$ consisted of $CsSnCl_3$, $SrRhO_3$ and $SrTiO_{3,}$ appeared to be an indirect bandgap through R-$\Gamma$ in a value of 1.05 eV. According to the graph, VBM was located on the R point but CBM was on the $\Gamma$ point. So the bandgap can barely reach the goal of 1.0 eV to 1.7 eV. The formation energy of the compound is rather low compared to previous combinations, which is 34.4. As the graph shows, the bands are likely to surround the Fermi level.

### 3.3.5. Implication

No previous literature has investigated or discussed perovskites mixtures. This paper exhibits a new method that demonstrates the possibility of retaining the stability and improving the efficiency of existing perovskites.

This paper also demonstrates the utility of adjusting bandgaps as a way of making unfit materials suitable for use. Finally, this paper explored the properties of mixed perovskites in hopes of contributing to their use in future applications.

## 4. Conclusions

In summary, we designed mixed perovskites to optimize bandgap and stability towards photovoltaic applications based on the understanding of the structural and electronic properties of halide-oxide perovskite compounds mixed in various configurations.

Upon stoichiometrically combining two different types of perovskites, all cubic perovskites generally exhibited dominant contributions on the CBM and VBM bands. The B-p and X-p states tended to be positioned at the top of the VBM and the bottom of the CBM respectively, producing B-x bonds in which the bonding intensity was dependent on the atomic radius. The hybridization between p-p bonds was generally weak. We next found that bandgaps could be readily adjusted via different configurations of unit cells. The compound $Cs_2Sn_2Cl_3I_3Sr_2TiRhO_6$ showed promise for photovoltaic applications, with bandgaps appearing at the $\Gamma$ point and within the ideal ranges of 1.0–1.7 eV solar cell energy applications. The compound $Cs_4Sn_4Cl_6I_6$ possessed an indirect bandgap of 1.9 eV as can be seen in Figure 5.

Negative formation energy value, according to the equation that we used to calculate it, indicates that the material is stable. Although $Cs_4Sn_4Cl_6I_6$ did not show a desirable bandgap value, its formation energy of 0.023 eV indicated high structural stability, further indicating advantages for use under harsh conditions. The two combinations can be applied under disparate conditions.

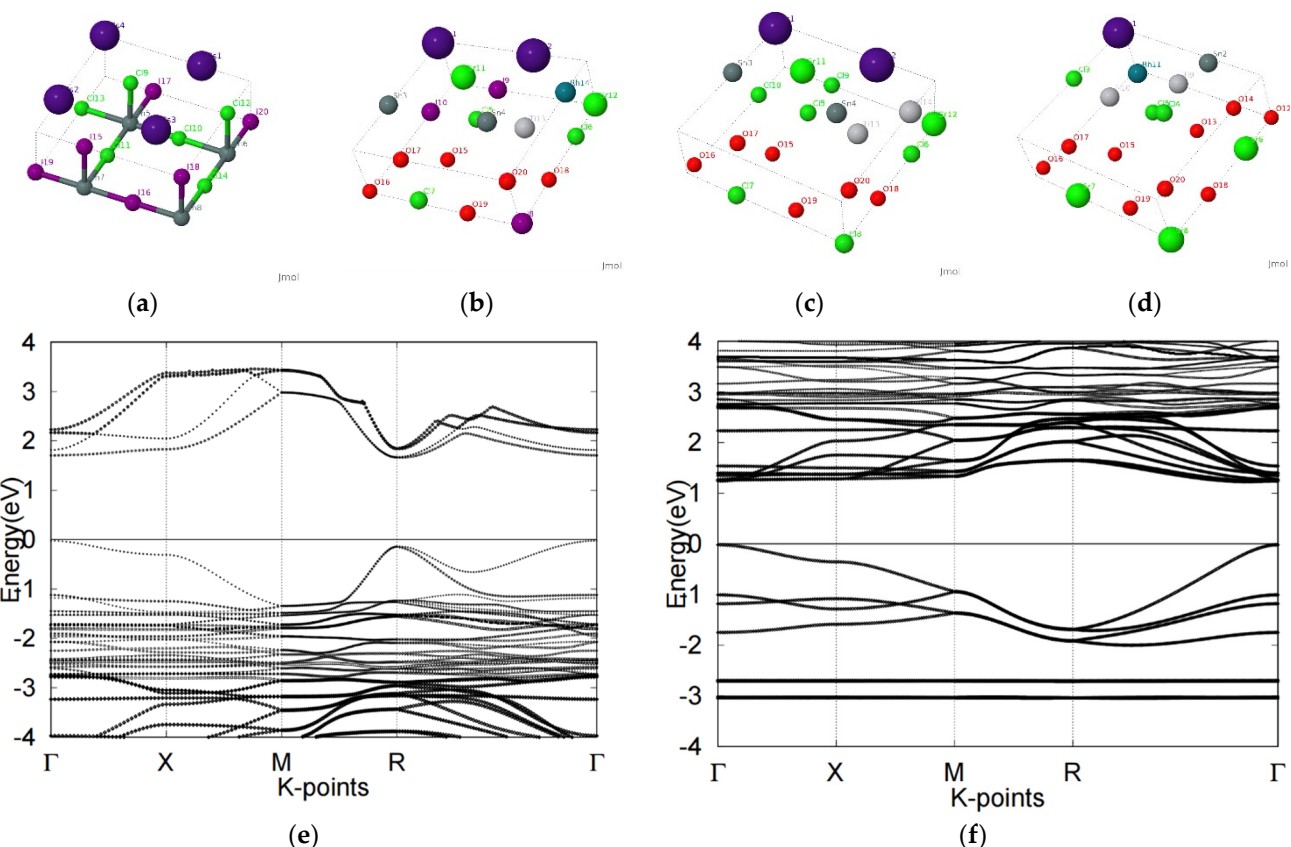

**Figure 5.** *Cont.*

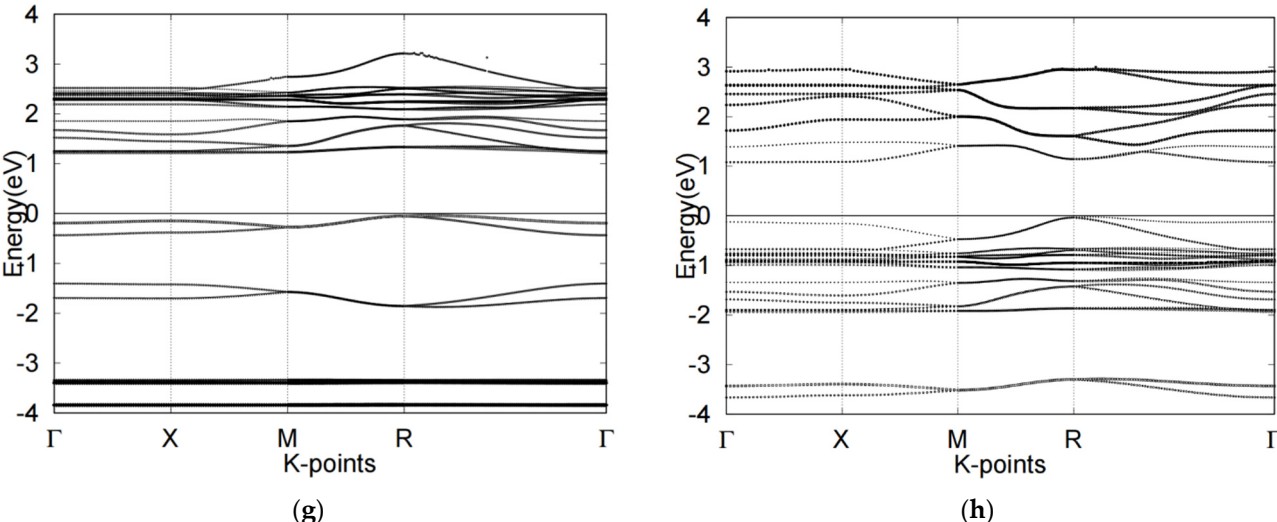

(**g**)  (**h**)

**Figure 5.** Atomic structures of (**a**) $Cs_4Sn_4Cl_6I_6$ (**b**) $Cs_2Sn_2Cl_3I_3Sr_2TiRhO_6$ (**c**) $Cs_2Sn_2Cl_6Sr_2Ti_2O_6$ and (**d**) $CsSnCl_3Sr_3Ti_2RhO_{12}$ mixed perovskite compounds were displayed along with the labels of the atoms. Electronic band structure of compounds (**e**) $Cs_4Sn_4Cl_6I_6$ (**f**) $Cs_2Sn_2Cl_3I_3Sr_2TiRhO_6$ (**g**) $Cs_2Sn_2Cl_6Sr_2Ti_2O_6$ (**h**) $CsSnCl_3Sr_3Ti_2RhO_{12}$ were also presented with various bandgaps.

Theoretical calculations do provide insight into the probable real-world trends to be found in the manipulation of these materials. However, it must be remembered that this theoretical calculation does not consider real-life factors. As such, experimental trial data and results may not be equivalent to those of theoretical calculations, even as they remain similar.

**Author Contributions:** Conceptualization, X.L. and H.W.; methodology, H.W.; software, X.L.; validation, H.W.; formal analysis, H.W.; investigation, H.W.; resources, H.W.; data curation, X.L.; writing—original draft preparation, H.W.; writing—review and editing, X.L.; visualization, H.W.; supervision, X.L.; project administration, X.L.; funding acquisition, H.W. Both authors have read and agreed to the published version of the manuscript.

**Funding:** This research received no external funding.

**Institutional Review Board Statement:** Not applicable.

**Informed Consent Statement:** Not applicable.

**Data Availability Statement:** No new data were created or analyzed in this study. Data sharing is not applicable to this article.

**Conflicts of Interest:** The authors declare that they have no conflict of interest.

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
