# Peer review of "Tuning Bandgaps of Mixed Halide and Oxide Perovskites CsSnX3 (X=Cl, I), and SrBO3 (B=Rh, Ti)"

_applsci, doi:10.3390/app11156862_

Round 1
Reviewer 1 Report
In this work, authors optimized mixed perovskite with a bandgap which is enough to harvest a large amount of photon from NIR-I region using first principles approach. I would recommend to accept the manuscript after addressing the following comments:
1) There is a scope to improve the abstract highlighting the major outcomes and potential use of mixed perovskite in the experimental setup. It will be helpful for reader if authors can mentioned the elements represents A and B.
2) In computational methods section, please fix the equation 1; GGA-PBE it maybe a subscript/superscript. Also mention the source of number used in the equation.
3) Please replace all Figure 3 and Figure 4 images with the higher resolution images, as the current version is so blur too tough to see the subtle changes.
Author Response
1) There is a scope to improve the abstract highlighting the major outcomes and potential use of mixed perovskite in the experimental setup. It will be helpful for readers if authors can mention the elements representing A and B.
In the section "pure materials," we added the elements representing A and B besides the title of the two minor sections "CsSnX" and "SrXO". For the main title of the paper, we already put the elements in parentheses beside it.
2) In the computational methods section, please fix the equation 1; GGA-PBE it may be a subscript/superscript. Also mention the source of the number used in the equation.
The problem about whether GGA-PBE in the equation should be a subscript/superscript, after I check with other similar papers online, I believe that it should not be either subscript/superscript. I compared the equation to a published paper, “Theoretical pressure-tuning bandgaps of double perovskites A2(BB’) X6 for photovoltaics,” and confirmed that there’s no problem with the equation.
3) Please replace all Figure 3 and Figure 4 images with the higher resolution images, as the current version is so blurry and too tough to see the subtle changes.
We replaced all of the blurry figures with higher resolution figures and adjusted the size of the figures in order to be clearly seen.
Reviewer 2 Report
The manuscript “Tuning bandgaps of mixed halide and oxide perovskites CsSnX3 (X=Cl, I), and SrBO3 (B=Rh, Ti)” by Hongzhe Wen and Xuan Luo deals with DFT calculations of band gap energies of four of four cubic “perovskites” CsSnCl3, CsSnI3, SrRhO3, and SrTiO3 and some other mixed ones
In general the manuscript is difficult to read, especially because no care was taken with its presentation. It is full of typos, separated words, gaps, and strange formatting, as well as small figures (as 3 and 4)
I personally do not trust these results, as authors say “As a result, underestimated bandgaps may appear in the final result of our cal-culation, and may therefore suggest higher efficiency than the bandgaps had when tuning the bandgap of mixed perovskites. In light of this and in order to render accurate data, we use Equation 1 to adjust the final value. Egap(corrected) = (Egap(GGA − PBE) × 1.358 + 0.125) × 0.998 + 0.014”
But not reasons are exposed for this correction.
In addition, no comparison is made with real materials, some of them widely studied, and bibliography appears necessary for these two aspects.
The manuscript needs a deep revision prior to be published in any journal.
Author Response
The manuscript “Tuning bandgaps of mixed halide and oxide perovskites CsSnX3 (X=Cl, I), and SrBO3 (B=Rh, Ti)” by Hongzhe Wen and Xuan Luo deals with DFT calculations of band gap energies of four of four cubic “perovskites” CsSnCl3, CsSnI3, SrRhO3, and SrTiO3 and some other mixed ones
In general the manuscript is difficult to read, especially because no care was taken with its presentation. It is full of typos, separated words, gaps, and strange formatting, as well as small figures (as 3 and 4)
We changed a few grammatical mistakes that appeared in the paper. Also, we substitute the figures with higher resolutions. The strange format is because of the transformation of the file format from pdf to docx. We fixed this problem as best as we could since a strange format can still appear when using different devices to open it.
I personally do not trust these results, as authors say “As a result, underestimated band gaps may appear in the final result of our cal-culation, and may therefore suggest higher efficiency than the band gaps had when tuning the bandgap of mixed perovskites. In light of this and in order to render accurate data, we use Equation 1 to adjust the final value. Egap(corrected) = (Egap(GGA − PBE) × 1.358 + 0.125) × 0.998 + 0.014”
But no reasons are exposed for this correction.
The results are all correctly calculated with the method mentioned in the paper. Reasons for using this equation to make corrections on the calculated result are written in the method section, around the part where the equation is located. Moreover, we provided the source where we got this equation.
In addition, no comparison is made with real materials, some of them widely studied, and bibliography appears necessary for these two aspects
It is current that some of the materials are widely studied, especially about the pure materials. Indeed, we compared our results with real experimental data and all data are included in Table III. However, as for the mixed materials, there’s no such information about the experimental data online or any articles. Due to this reason, there’s no way that we can confirm it unless we perform a further study on these materials.
The manuscript needs a deep revision prior to being published in any journal.

Round 2
Reviewer 2 Report
Authors have improved the presentation of their results, and some key aspects have been solved, so it is more readable, but despite not being my cup of tea, results appear acceptable, although it is dissapointing the lack of comparison for mixed perovskites.
However, I think that some aspects, as separated words should be revised, as I cannot see the need of so many ones, and some of them are actually strange.
Author Response
However, I think that some aspects, as separated words should be revised, as I cannot see the need of so many ones, and some of them are actually strange.
We are uncertain about the exact meaning of this comment. So we had deleted some misused hyphens in order to make the taxt clearer. This may caused by the trasnformation from pdf to docx as well.
